# The Role of Urotensin-II in Obesity and Metabolic Syndrome in Pediatric Population

**DOI:** 10.3390/children9020204

**Published:** 2022-02-04

**Authors:** Marko Simunovic, Andrija Jukic, Martina Paradzik, Daniela Supe-Domic, Lada Stanisic, Marina Degoricija, Anna Hummelvoll Hillestad, Veselin Skrabic, Josko Bozic

**Affiliations:** 1Department of Pediatrics, University Hospital of Split, Spinciceva 1, 21000 Split, Croatia; vskrabic@kbsplit.hr; 2Department of Pediatrics, University of Split School of Medicine, Soltanska 2, 21000 Split, Croatia; 3Institute of Emergency Medicine in Split-Dalmatia County, Spinciceva 1, 21000 Split, Croatia; andrija.jukic94@gmail.com; 4Department of Ophthalmology, University Hospital of Split, Spinciceva 1, 21000 Split, Croatia; martina.paradzik@gmail.com; 5Department of Medical Laboratory Diagnostics, University Hospital of Split, Spinciceva 1, 21000 Split, Croatia; daniela.supedomic@gmail.com (D.S.-D.); lada.stanisic@kbsplit.hr (L.S.); 6Department of Health Studies, University of Split, Rudjera Boskovica 35, P.P. 464, 21000 Split, Croatia; 7Department of Medical Chemistry and Biochemistry, University of Split School of Medicine, Soltanska 2, 21000 Split, Croatia; marina.degoricja@mefst.hr; 8Internal Medicine Ward, Sørlandet Hospital Flekkefjord, Engvald Hansens vei 6, 4400 Flekkefjord, Norway; anna.hillestad@gmail.com; 9Department of Pathophysiology, University of Split School of Medicine, Soltanska 2, 21000 Split, Croatia; josko.bozic@mefst.hr

**Keywords:** urotensin-II, obesity, metabolic syndrome, hypertension, children, adolescents

## Abstract

Background: Urotensin-II (U-II) is a short cyclic peptide that is widely recognized as one of the most potent vasoconstrictors. U-II plays a role in the pathophysiology of MS, participating in the development of essential hypertension, insulin resistance, hyperglycemia, and a proinflammatory state. Methods: This study comprised 52 obese children and adolescents with a body mass index (BMI) z score > 2, aged 10 to 18 years. Serum levels of U-II were assessed using an enzyme-linked immunosorbent assay along with other standard biochemical parameters. Results: Elevated serum levels of U-II were recorded in the group of obese subjects with MS when compared with the group of obese subjects without MS (4.99 (8.97–3.16) vs. 4.17 (5.17–2.03) ng/mL, median and IQR, *p* = 0.026). Furthermore, a subgroup of study subjects with high blood pressure had significantly higher U-II levels in comparison with the normotensive subgroup (4.98 (7.19–3.22) vs. 3.32 (5.06–1.97) ng/mL, *p* = 0.027), while the subgroup with a positive family history of high blood pressure had significantly higher U-II levels when compared with subjects who had a negative family history of elevated blood pressure (5.06 (6.83–4.45) vs. 3.32 (6.13–2.21) ng/mL, *p* = 0.039). Conclusions: To the best of the author’s knowledge, this is the first study on the levels of U-II in obese children and adolescents, including a possible link to MS.

## 1. Introduction

Pediatric obesity has become one of the most important public health concerns in countries across the world [1,2,3,4]. In the United States, around 30 percent of children and adolescents are overweight or obese, while obesity prevalence in Europe varies from 5% to 31% depending on country reports [1,5]. As a result of this, various major obesity-related comorbidities have emerged [6]. Obesity in children often persists into adulthood and is significantly linked to type 2 diabetes with a high risk for cardiovascular disease later in life [7,8,9]. Many of the metabolic and cardiovascular complications of obesity begin in childhood and are directly linked to the occurrence of insulin resistance and hyperinsulinemia, the most frequent metabolic abnormality linked to obesity [10]. Insulin resistance plays a central role in the pathophysiology underlying the metabolic syndrome (MS), which is defined as a collection of risk factors including hypertension, impaired glucose metabolism, elevated triglycerides, and low high-density lipoprotein (HDL) cholesterol as well as abdominal obesity [11,12,13,14,15]. MS is associated with endothelial dysfunction combined with proinflammatory and prothrombotic states, which are independent risk factors for type 2 diabetes and cardiovascular disease [13,16].

Urotensin-II (U-II) is a short cyclic peptide that was discovered in the spinal cord of the teleost fish in 1980 and it is considered to be one the most powerful vasoconstrictors [17,18,19,20]. It is produced by the cells of the blood vessels, heart, kidneys and liver, but it is also found in the central nervous system as well as the gastrointestinal tract [21,22]. Several studies on the cardiovascular effects of U-II have found that increased plasma U-II levels are associated with diabetes, hypertension, and renal failure, but also a variety of cardiovascular illnesses, including congestive heart failure and carotid atherosclerosis [23,24,25,26]. Polymorphisms in the U-II gene have been related to diabetes and insulin resistance [19]. Moreover, studies suggest that the U-II plays a role in the pathophysiology of MS, participating in the development of essential hypertension, insulin resistance, hyperglycemia, and proinflammatory state [27]. Therefore, our study aimed to compare the serum levels of urotensin II between obese children and adolescents with and without MS. The secondary aim of this study was to investigate the possible relationship between U-II levels and other cardiovascular risk factors.

## 2. Materials and Methods

Study participants comprised 52 obese children and adolescents aged 10 to 18 years with a body mass index (BMI) z score > 2, who were recruited from April 2017 to June 2018 at the Department of Pediatric Endocrinology of the University Hospital of Split [3,28]. All the procedures in this study were performed following the ethical principles regarding human experimentation defined by the Declaration of Helsinki and the study received approval from the University Hospital of Split, Ethical Committee (2181-147-01/06/M.S.-17-2, 500-3/17-01/10). Exclusion criteria in this study included monogenic and secondary causes of obesity, diseases and disorders, and medicaments that affect hypertension, glucose, and lipid metabolism. Each child gave verbal approval for the study and informed consent was disclosed and signed by the parent.

### 2.1. Anthropometric and Clinical Assessments

A standard protocol for clinical assessment was used for all subjects enrolled in our study as described previously [3,28]. Body height, weight, and waist circumference (WC) were measured and BMI was calculated for all subjects. The stage of puberty was determined according to the Tanner scale [29,30]. Furthermore, blood pressure was measured according to the standard protocol from the Task Force for high blood pressure in children and adolescents, and detailed family history was obtained for a positive family history of high blood pressure [31,32].

### 2.2. Definitions

BMI z score was calculated using World Health Organization growth charts adjusted for age and sex (AnthroPlus software, WHO, Geneva, Switzerland) and the threshold for obesity was BMI z score > 2 [33,34,35].

The 2007 International Diabetes Federation (IDF) pediatric definition was used for diagnosing metabolic syndrome [11].

Homeostatic model assessment of insulin resistance (HOMA-IR) was calculated for all patients [36]. In addition, a positive family history of high blood pressure was established when the subject’s parents had a diagnosis of hypertension or had hypertension controlled by antihypertensive drugs, or both [3,37].

### 2.3. Laboratory Analysis

Blood was sampled from subjects after at least 12 h of overnight fasting. All procedures were conducted by an experienced medical biochemist in a blinded manner. The levels of serum U-II were determined by an enzyme-linked immunosorbent assay (ELIA kit, Phoenix Pharmaceuticals Inc., Burlingame, CA, USA), following the manufacturer’s instructions. The sensitivity of the assay was 0.06 ng/mL and the linear range was 0.06 to 8.2 ng/mL. The intra-assay coefficient of variation (CV) was less than 10% and the inter-assay CV was less than 15%. Furthermore, all the other biochemical analyses were performed following good laboratory standard practice.

### 2.4. Data Analysis

Data analysis was performed using MedCalc for Windows and Prism 9 for Mac OS. Categorical variables were presented as whole numbers (N) and percentages (%), while continuous data were expressed as mean ± standard deviation or median and interquartile range. Kolmogorov–Smirnov test was used to assess the normal distribution of data. Student t-test or Mann–Whitney U tests were used for comparison of continuous data, whereas for categorical data, Chi-squared test was employed. Furthermore, the independent predictor of having MS was valuated with multivariable logistic regression, with odds ratio, 95% confidence interval, and *p* value. The statistical significance level was set to *p* < 0.05.

In a pilot study that was performed with 10 randomly selected obese study subjects with MS and 10 obese study subjects without MS, U-II levels were assessed. The difference between the two means of U-II levels was 1.1 and SD was 1.3. The power was set to 80%, considered type I error was set to 0.05, and the calculated minimum sample size was 22 subjects per group.

## 3. Results

The 52 patients included in the study were divided into groups of 26 subjects depending on the presence of MS. Obese subjects with MS had mean age 16.42 ± 8.34 years whereas gender distribution was 13 (50%) male and 13 (50%) females, while in the group of obese subjects without MS the average age was 14.12 ± 2.27 years and gender distribution was 18 (69.2%) male and 13 (30.8%) females. Furthermore, there were no statistically significant differences in height, weight, BMI, BMI z score, waist circumference, or pubertal stage between groups (Table 1).

The group of obese subjects with MS had significantly higher levels of triglycerides (1.51 ± 0.69 vs. 1.02 ± 0.38 mmol/L, *p* = 0.003), systolic blood pressure (135.5 ± 15.56 vs. 124 ± 10.01 mmHg, *p* = 0.003), and significantly lower levels of HDL cholesterol (1.00 (1.00–0.90) vs. 1.20 (1.30–1.10) mmol/L, *p* < 0.001) when compared with group of obese subjects without MS. Additional comparisons of biochemical characteristics between the two groups are presented in Table 2.

Serum levels of U-II were significantly higher in the group of obese subjects with MS in comparison with the group of obese subjects without MS (4.99 (8.97–3.16) vs. 4.17 (5.17–2.03) ng/mL, *p* = 0.026), as presented in Figure 1. There was no significant difference in U-II levels between male and female subjects (4.63 (6.86–2.42) vs. 4.52 (6.47–2.84), *p* = 0.740).

Mann–Whitney U test was performed to compare the U-II levels between groups. Data are presented as a median and interquartile range, while significance (*p*) was set to <0.05.

Additionally, logistic regression analysis, after adjusting for age, sex, a positive family history of high blood pressure, and BMI, revealed that elevated U-II levels were positively associated with a risk factor for the development of MS (OR 1.38, 95% confidence interval 1.01–1.92, *p* = 0.046), as presented in Table 3. 

Additional stratification of the study population was performed according to the presence of elevated systolic and diastolic blood pressure. The subgroup of 32 subjects with high blood pressure had significantly higher U-II levels when compared to the normotensive subgroup (4.98 (7.19–3.22) vs. 3.32 (5.06–1.97) ng/mL, *p* = 0.027) (Figure 2). 

A subgroup with elevated systolic and diastolic blood pressure consisted of 32 subjects. Mann–Whitney U test was performed to compare serum U-II levels between groups. Data are presented as a median and interquartile range, while significance (*p*) was set to <0.05.

Furthermore, a subgroup of 21 subjects with a positive family history of high blood pressure had significantly higher U-II levels in comparison with study subjects with a negative family history of elevated blood pressure (5.06 (6.83–4.45) vs. 3.32 (6.13–2.21) ng/mL, *p* = 0.039) (Figure 3).

A subgroup with a positive family history of high blood pressure consisted of 21 subjects. Mann–Whitney U test was performed to compare the U-II levels between groups. Data are presented as a median and interquartile range, while significance (*p*) was set to <0.05.

## 4. Discussion

In this cross-sectional study, we reported that obese children and adolescents with MS had significantly higher serum levels of U-II in comparison with children and adolescents without MS. Additionally, we demonstrated that obese children and adolescents with high blood pressure or positive family history of high blood pressure had significantly higher serum levels of U-II. To the best of the author’s knowledge, this is the first study on the levels of U-II in obese children and adolescents, including a possible link to MS.

A study on the experimental model of obese mice has demonstrated a possible link between U-II and MS including the cardiovascular risk factors [38]. Blocking the U-II receptor pathway in obese mice had a positive effect on both weight reductions and components of MS [38]. Furthermore, U-II receptors were found in the hypothalamus of the experimental model and U-II may affect the complex appetite-regulating network [27,39]. In addition, U-II affects lipid metabolism in the liver influencing lipogenesis via the glucose-6-phosphate dehydrogenase pathway and nicotinamide adenine dinucleotide phosphate (NADPH) activity [22,27]. In a recent study on resistant hypertensive adults, U-II levels significantly correlate with HDL level (r = 0.306, *p* = 0.031) [40].

Several studies connect U-II with the pathophysiological cascade of atherosclerosis and low-grade inflammation, which is one of the leading pathological pathways in the development of MS [3,24,27,41]. In a recent study in patients with inflammatory bowel disease, a positive correlation between U-II and high sensitivity C-reactive protein, a well-established biomarker of low-grade inflammation, has been observed [42]. Several studies demonstrated a possible association of U-II with indices of insulin resistance, hyperinsulinemia, and impaired glucose tolerance in hypertensive patients [3,27,43]. In addition, the relationship of U-II and insulin resistance was demonstrated in the study by Demirpence et al. in a population of adult patients with acromegaly where U-II levels were positively correlated with HOMA-IR (r = 0.231, *p* = 0.033), which is one of the most frequently used methods for assessing insulin resistance [3,44]. In this regard, it is well known that insulin resistance and low-grade inflammation are recognized as the most important factors in the development of MS [3,27,43]. Moreover, after long-term administration of U-II to high-fat diet-fed mice, there was a significant improvement in glucose metabolism, which led to a reduction of adipose tissue and weight [45]. Another recent cohort study on the adult population showed that mutation rs2890565 of the U-II gene is associated with increased risk for type 2 diabetes [46]. Moreover, in a study by Cheung et al. conducted on a hypertensive adult population, a positive correlation between U-II levels and weight was demonstrated [23]. All of this suggests an important role of U-II in a complex pathophysiological mechanism underlying the progression of MS. Moreover, our study further strengthens the possible link between U-II and MS at the onset and progression of cardiovascular complications in the pediatric population, providing a rationale for longitudinal studies in the obese population of children and adolescents, which would monitor the long-term effects of U-II on the progression of cardiovascular complications and the development of MS.

Another major finding of the present study is a possible connection between U-II and hypertension. It was previously established that U-II is one of the most potent vasoconstrictors contributing to increased peripheral resistance but also exerting effects on cardiac function [17]. These findings have been confirmed in several studies in the adult population where a clear association between U-II and hypertension has been shown [23,40,41,42,47]. A genetic study confirmed that S89N single-nucleotide polymorphism of the U-II gene plays a key role in the development of essential hypertension which possibly explains the independent association of elevated U-II levels in obese subjects with a positive family history of high blood pressure in our study [48]. Furthermore, in one pediatric study with subjects with chronic liver disease and portal hypertension, U-II was significantly higher compared with healthy controls, indicating that U-II effects the liver metabolism, but further studies are needed to clarify the role of U-II in non-alcoholic fatty liver disease in the obese pediatric population [49].

There are several limitations to our study. First, the cross-sectional single-center design of the study prevents us from making any causal inferences. Second, the sample size of the study could affect the reproducibility of the results. In addition, there is a partial overlap between subgroups, which could affect our results. Finally, the inability to include children under the age of 10, due to the uncertain definition of MS in young children, might also affect our results.

## 5. Conclusions

In conclusion, obese children and adolescents with MS had higher serum levels of U-II in comparison with obese pediatric patients without MS. Based on these findings, U-II represents a potential new research target in the pathophysiology of MS in the pediatric population. Nevertheless, future studies are needed to define the role of U-II in the onset and progression of MS, as well as to elucidate the clinical implications of U-II in MS.

## Figures and Tables

**Figure 1 children-09-00204-f001:**
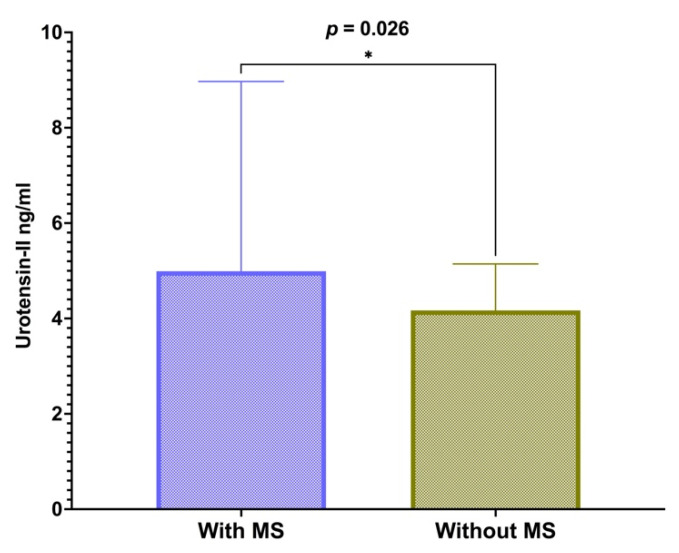
Levels of serum urotensin-II regarding the presence of the metabolic syndrome. * Significance level (*p*) was set to <0.05.

**Figure 2 children-09-00204-f002:**
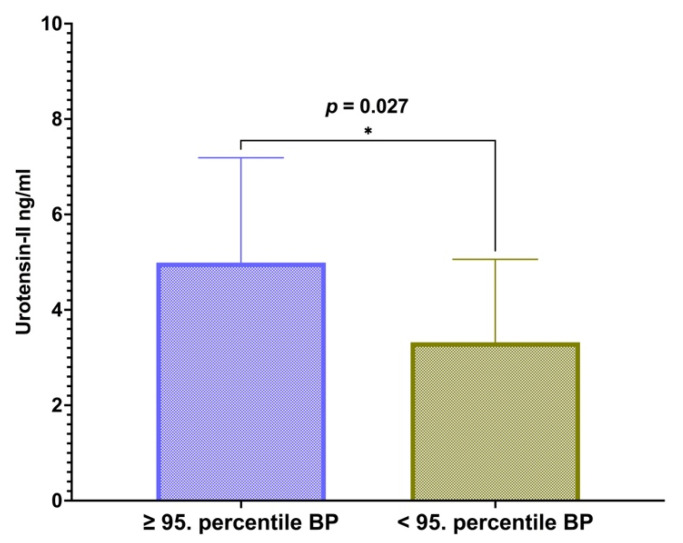
Association of urotensin-II with elevated systolic and diastolic blood pressure in obese children and adolescents. * Significance level (*p*) was set to <0.05.

**Figure 3 children-09-00204-f003:**
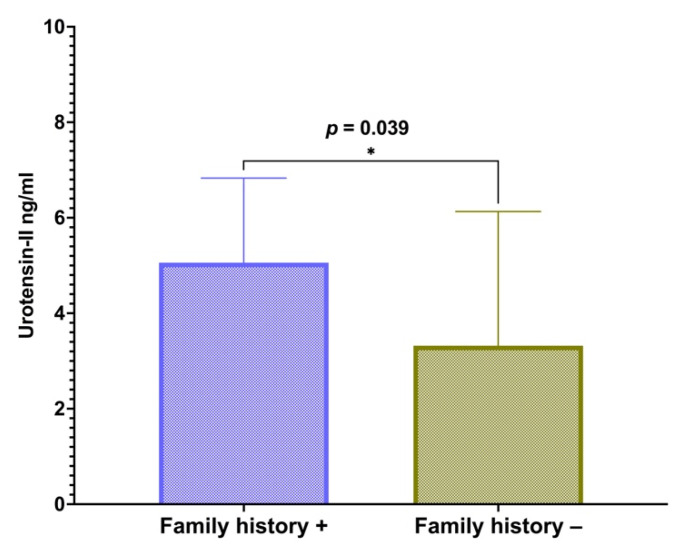
Association of urotensin with a positive family history of high blood pressure in obese children and adolescents. * Significance level (*p*) was set to <0.05.

**Table 1 children-09-00204-t001:** Anthropometric characteristics of the research subjects.

Factor	With Metabolic Syndrome (*n* = 26)	Without Metabolic Syndrome (*n* = 26)	*p* *
Gender–N (%)			
Male	13 (50)	18 (69.2)	0.158
Female	13 (50)	8 (30.8)	
Age (years)	16.42 ± 8.34	14.12 ± 2.27	0.172
Height (cm)	170.0 ± 11.72	167.6 ± 12.98	0.476
Weight (kg)	91.34 ± 17.74	87.54 ± 18.84	0.458
BMI (kg/m^2^)	31.44 ± 3.92	30.7 ± 3.62	0.456
BMI z score	2.82 ± 0.56	2.76 ± 0.43	0.661
Waist circumference	103.6 ± 9.25	102.3 ± 9.19	0.617
Pubertal status—N (%)			
Tanner 1	2 (7.7)	4 (15.4)	0.793
Tanner 2	3 (11.5)	2 (7.7)	
Tanner 3	4 (15.4)	4 (15.4)	
Tanner 4	8 (30.8)	10 (38.4)	
Tanner 5	9 (34.6)	6 (23.1)	

MS, metabolic syndrome; BMI, body mass index. The differences between study group anthropometric characteristics were tested using the Student’s *t* test or χ2 test. * Significance level (*p*) was set to <0.05.

**Table 2 children-09-00204-t002:** Biochemical characteristics of the research subjects.

Factor	With Metabolic Syndrome (*n* = 26)	Without Metabolic Syndorme (*n* = 26)	*p* *
Total cholesterol (mmol/L)	4.4 (4.80–3.65)	4 (4.9–3.78)	0.487
Triglycerides (mmol/L)	1.51 ± 0.69	1.02 ± 0.38	0.003
HDL cholesterol (mmol/L)	1.00 (1.00–0.90)	1.20 (1.30–1.10)	<0.001
LDL cholesterol (mmol/L)	2.70 (3.13–2.20)	2.50 (3.03–2.00)	0.190
Fasting glucose (mmol/L)	5.09 (5.45–4.81)	4.98 (5.3–4.79)	0.370
Fasting insulin (mIU/L)	25.7 (42.23–19.75)	23.6 (37.23–15.98)	0.315
HOMA-IR	5.97 (10.02–4.30)	5.56 (7.99–3.55)	0.351
HemoglobinA1c (%)	5.49 ± 0.36	5.43 ± 0.31	0.512
SBP (mmHg)	135.5 ± 15.56	124.0 ± 10.01	0.003
DBP (mmHg)	79.50 (85.00–75.00)	75.00 (80.50–70.00)	0.120

MS, metabolic syndrome; HDL cholesterol, high-density lipoprotein cholesterol; LDL cholesterol, low-density lipoprotein cholesterol; HOMA-IR, homeostasis model assessment of insulin resistance index; SBP, systolic blood pressure; DBP, diastolic blood pressure. The differences between study group characteristics were tested using the Student’s *t* test or Mann–Whitney test. * Significance level (*p*) was set to <0.05.

**Table 3 children-09-00204-t003:** Multivariable logistic regression model of prediction for metabolic syndrome.

Factor	Odds Ratio	95% CI	*p* *
Age (years)	0.73	0.48–1.09	0.142
Sex	3.59	0.77–16.74	0.104
BMI (kg/m^2^)	1.20	0.96–1.50	0.102
Urotensin-II (ng/mL)	1.38	1.01–1.92	0.046
Positive family history	3.92	0.79–19.43	0.095

BMI, body mass index. Multivariable logistic regression, with odds ratio, 95% confidence interval. * Significance level (*p*) set to <0.05.

## Data Availability

The data presented in this study are available on request from the corresponding author. The data are not publicly available due to restrictions e.g., privacy or ethics.

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
