# Peer review of "The Role of Urotensin-II in Obesity and Metabolic Syndrome in Pediatric Population"

_children, 2022, doi:10.3390/children9020204_

Round 1

Reviewer 1 Report

The manuscript examines levels of U-II in obese children with and without MS. It has the potential to provide interesting insight into the pathophysiology of juvenile MS. However, only fairly rudimentary analysis are performed, specifically there seems to be only one result  the correlation between U-II and hypertension or history of hypertension could be interesting, but that would only be so if there was not a significant overlap between the three groups and this is not reported. Additionally, there is no evidence to suggest that U-II has any role in the onset of MS, or if it is a result of MS, or if the correlation is significant in any clinical way - more investigation is needed. Lastly, more information should be added to better describe how U-II may have a mechanistic role in MS, as is implied.

[lines 22-23] More information is needed on the methods in the abstracts, such as what was measured and if any intervention took place. 

[lines 24-29] It would be expected that there would be significant overlap between these groups (MS, hypertensive and family history of hypertension), especially between the last two, so it seems hardly surprising that they all share the same biochemical marker. Was the overlap between these groups calculated?

[abstract] It should be clarified what the statistics given are, e.g. if they are mean and 95% confidence interval.

[Table 1] Were the male and female data combined? If so, what was the rationale for doing so?

[lines 149-152] As increased U-II was already shown to be correlated with MS, what is the significance of it also correlating with risk factors for MS? By definition these risk factors would directly correlate to the presence of MS so wouldn’t the related correlation to elevated U-II be derivative?

[lines 154-155] The legend for Figure 1 needs to be corrected. 

[lines 156-160 and 164-167] As above: was the overlap between these stratification and the original MS groupings investigated?

[figures] The figure legends need more information, such as sample size and description of how the data is displayed (e.g. mean +/- SEM)

[lines 200-201] Firstly, how is this a major finding if this has already been shown in multiple adult studies? Secondly, as hypertension is a part of MS it seems obvious that hypertension would also correlated to increased U-II levels, unless this correlation was only calculated among participants without MS?

Author Response

Thank you very much for considering our manuscript for possible publication in the Children. We have put in all our efforts to address every question and comment provided. We are thankful for these constructive comments which have improved our manuscript.

 We would like to address the following comments of Reviewer 1:

  1. We would like to address the following comment:

 “The manuscript examines levels of U-II in obese children with and without MS. It has the potential to provide interesting insight into the pathophysiology of juvenile MS. However, only fairly rudimentary analysis are performed, specifically there seems to be only one result the correlation between U-II and hypertension or history of hypertension could be interesting, but that would only be so if there was not a significant overlap between the three groups and this is not reported. Additionally, there is no evidence to suggest that U-II has any role in the onset of MS, or if it is a result of MS, or if the correlation is significant in any clinical way - more investigation is needed. Lastly, more information should be added to better describe how U-II may have a mechanistic role in MS, as is implied.

Comment: Following the Reviewer’s comment, we would like to point out that the hypertensive subgroup consisted of obese patients with MS 21 (65 %) and without MS 15 (35 %), P=0.07, whereas the subgroup with a positive family history of hypertension consisted of obese patients with MS 13 (61 %) and without MS 8 (39 %), P=0.158. Furthermore, when comparing the subgroups, the subgroup of patients with high blood pressure, overlapped by 15 (46 %) patients from a subgroup of patients with a positive family history of high blood pressure. In accordance with this, we expanded the section concerning the limitations of the study with the possible influence of overlap of subgroups on results interpretation. Furthermore, U-II has previously been linked with abdominal obesity, which is the principal clinical risk factor for the development of MS. (ref. 27) MS is a cluster of metabolic disorders in which insulin resistance and low-grade inflammation play a major role in the pathophysiological mechanism. Several previous studies have demonstrated the link between U-II and low-grade inflammation and insulin resistance. In our study, we hypothesized that U-II is a new component in the complex pathophysiological mechanism leading to the onset of MS in the pediatric population. Our study reinforces the plausible thesis that U-II plays a distinct role during the onset of MS, considering that our studied group had an average age of 16 years which positions the studied population at the onset of cardiovascular complications. However, the mechanism of U-II actions during the onset of MS is beyond the scope of this study and further studies are needed to fully clarify the role U-II in MS. To further address and elaborate the background and plausibility for the mechanistic role of U-II in MS, we have now expanded the Discussion section.

  1. We would like to address the following comment:

[lines 22-23] More information is needed on the methods in the abstracts, such as what was measured and if any intervention took place.

Comment: According to the reviewer’s suggestion, we have further expanded the methods in the Abstract section to clarify which intervention took place.

  1. We would like to address the following comment:

“[lines 24-29] It would be expected that there would be significant overlap between these groups (MS, hypertensive and family history of hypertension), especially between the last two, so it seems hardly surprising that they all share the same biochemical marker. Was the overlap between these groups calculated?”

Comment: Once again we would like to express gratitude for the reviewer extensive revision and we addressed this comment in the previous section (part 1) discussing the apparent overlap in the Manuscript.

  1. We would like to address the following comment:

“[abstract] It should be clarified what the statistics given are, e.g. if they are mean and 95% confidence interval.

Comment: According to the Reviewer’s suggestions, we clearly stated what statistics was given for the results in the abstract.

  1. We would like to address the following comment:

“[Table 1] Were the male and female data combined? If so, what was the rationale for doing so?

Comment: We would like to point out that our study group with MS and without MS did not have statistically significant differences in gender distribution (male 13 (50 %) vs. 18 (69.2 %), female 13 (50 %) vs. 8 (30.8 %), P=0.158) and age (16.42 ± 8.34 vs. 14.12 ± 2.27, P=0.172). Furthermore, we paid particular attention to puberty status to rule out the possible impact of puberty on our results. To address this question, an additional analysis was performed to demonstrate that there were no significant differences in U-II levels between male and female subjects (4.63 (6.86-2.42) vs. 4.52 (6.47-2.84), p = 0.740). and we added this result to the Results section to further clarify the relationship between U-II levels and gender.

  1. We would like to address the following comment:

[lines 149-152] As increased U-II was already shown to be correlated with MS, what is the significance of it also correlating with risk factors for MS? By definition these risk factors would directly correlate to the presence of MS so wouldn’t the related correlation to elevated U-II be derivative?

Comment: Following the Reviewer’s comment, we would like to point out that we performed logistic regression analysis with adjustment for age and BMI. Our study group aged 10 to 18 years is heterogeneous and completely different from the typical adult cohort because puberty occurs and during this period various changes occur in the adolescent body and linear growth accelerates and this might affect the U-II levels. Furthermore, obesity is a fundamental component of MS and we tried to assess the independent impact of U-II on MS regardless of the degree of obesity. We further expanded the Discussion clarifying the relationship between U-II levels and MS.

  1. We would like to address the following comment:

[lines 154-155] The legend for Figure 1 needs to be corrected.

Comment: We modified the Figure 1 legend to better explain its contents.

  1. We would like to address the following comment:

[lines 156-160 and 164-167] As above: was the overlap between these stratification and the original MS groupings investigated?

Comment: We responded to this comment in the previous section by discussing the overlap in the Manuscript. (part 1)

  1. We would like to address the following comment:

[figures] The figure legends need more information, such as sample size and description of how the data is displayed (e.g. mean +/- SEM)

Comment: We modified all Figures as the Reviewer suggested so that the data in the figures would be more clearly presented.

  1. We would like to address the following comment:

[lines 200-201] Firstly, how is this a major finding if this has already been shown in multiple adult studies? Secondly, as hypertension is a part of MS it seems obvious that hypertension would also correlated to increased U-II levels, unless this correlation was only calculated among participants without MS?

Comment: Firstly, we would like to point out that this is the first study on the levels of U-II in obese children and adolescents which demonstrates a possible direct link to the onset of MS in the pediatric population. Pediatric obesity is one of the leading public health problems today. Obese children and adolescents have a significant prospect of becoming obese adults. Furthermore, the prevalence of MS is extremely high in the obese pediatric population. Our study is the first to compare the levels U-II between groups of obese pediatric patients with and without MS. Previous studies in the adult population were addressing the link between U-II and certain individual components of MS therefore conclusions were deducted about a possible link between MS and U-II in the adult population. Secondly, studies in adults demonstrated the association between U-II and an essential hypertension without the presence of MS. We further expanded the Discussion section to better explain the relationship between U-II levels and MS.

Reviewer 2 Report

Dear editor, first of all, I would like to thank you for the opportunity to review this work, which seems to me to be of great interest as it presents the possibility of a new parameter, urotensin II, to define the risk of metabolic syndrome in adolescents.
The methods used are adequate for the objective outlined, but in my opinion, some changes should be made, thus, it would be necessary to establish the population on which the study has been carried out and how the sample has been chosen, to know if it would be expandable. In the same sense, it would be necessary to specify if the sample size has been calculated or if it has been done to all those who met the inclusion criteria during a period of time, etc.
In the results, it is necessary to take care of some presentation details, thus, both in the text and in the tables, the same decimal figures must be placed in means and standard deviations and the p with lowercase letters.
Finally, a table should be placed with the results of the multivariate regression analysis and complemented with a binary logistic regression analysis with respect to SD (yes / no), excluding the parameters included in its definition, in order to calculate the risk of factors such as a family history of hypertension and the value of Urotensin II.

Author Response

Thank you very much for considering our manuscript for possible publication in the Children. We have put in all our efforts to address every question and comment provided. We are thankful for these constructive comments which have improved our manuscript.

  1. We would like to address the following comment:

“The methods used are adequate for the objective outlined, but in my opinion, some changes should be made, thus, it would be necessary to establish the population on which the study has been carried out and how the sample has been chosen, to know if it would be expandable. In the same sense, it would be necessary to specify if the sample size has been calculated or if it has been done to all those who met the inclusion criteria during a period of time, etc.”

Comment: Once again we would like to express gratitude for the Reviewer's extensive revision and we would like to apologize for not clearly stating how we sampled the patients in our study. Subjects were recruited in the University Hospital of Split, Division of Pediatric Endocrinology from April 2017 to June 2018. In our study, 52 obese subjects were randomly enrolled to meet the equal number of subjects in both groups with and without MS. A pilot study that included 10 subjects in both groups was conducted to calculate the sample size for our study. We further expanded the Methods section to better define how the sampling procedure was performed for the study population.

  1. We would like to address the following comment:

In the results, it is necessary to take care of some presentation details, thus, both in the text and in the tables, the same decimal figures must be placed in means and standard deviations and the p with lowercase letters.

Comment: According to the Reviewer’s suggestions, we improved the Results section and Tables to be in a consistent form.

  1. We would like to address the following comment:

Finally, a table should be placed with the results of the multivariate regression analysis and complemented with a binary logistic regression analysis with respect to SD (yes / no), excluding the parameters included in its definition, in order to calculate the risk of factors such as a family history of hypertension and the value of Urotensin II.

Comment: According to the Reviewer’s suggestions we created Table 3 with results of the multivariate regression and logistic regression analysis.

Round 2

Reviewer 1 Report

Most of my concerns have been sufficiently addressed.

Were other analyses conducted to confirm there was no sex differences? For example, U-II levels in subjects with and without MS separately for both sexes. In the group without MS there is over twice as many males as females, potentially skewing the data if the two sexes respond differently.

The legend for figure 1 is still in need of correction.

Author Response

  1. We would like to address the following comment:

 “Were other analyses conducted to confirm there was no sex differences? For example, U-II levels in subjects with and without MS separately for both sexes. In the group without MS there is over twice as many males as females, potentially skewing the data if the two sexes respond differently.

Comment: Following the Reviewer’s comment, we would like to point out that the prevalence of overweight and obese children in our school-age population is 42.1 % in boys and 32.1 % in girls (Musić Milanović S et al. Acta Clin Croat.2020;59(2):303-311). Considering these gender differences in the prevalence of the overall population, a difference in the distribution between the genders in our study group is expected. In addition to this, we would like to emphasize that at the moment, there are no differences in the approach in diagnosis and treatment between genders in the obese pediatric population. To address the Reviewers comment, an additional analysis was performed which demonstrated that there were no significant differences in serum U-II levels between male and female subjects in the group of obese subjects with MS (6.86 (8.97-3.76) vs. 4.64 (9.34-2.91), p = 0.405) and in the group of obese subjects without MS 4.27 (5.19-2.03) vs. 3.32 (5.17-1.66), p = 0.802).

  1. We would like to address the following comment:

“The legend for figure 1 is still in need of correction.”

Comment: Thank you for the remark, the legend of Figure 1 is now corrected.